# Position: Modular Memory is the Key to Continual Learning Agents

Vaggelis Dorovatas[1]   Malte Schwerin[2]   Andrew D. Bagdanov[3]   Lucas Caccia[4]   Antonio Carta[5]
Laurent Charlin[6]   Barbara Hammer[7]   Tyler L. Hayes[8]   Timm Hess[9]   Christopher Kanan[10]
Dhireesha Kudithipudi[11]   Xialei Liu[12]   Vincenzo Lomonaco[13]   Jorge Mendez-Mendez[14]   Darshan Patil[6]
Ameya Prabhu[15]   Elisa Ricci[16]   Tinne Tuytelaars[9]   Gido M. van de Ven[17]   Liyuan Wang[18]
Joost van de Weijer[19]   Jonghyun Choi[20]   Martin Mundt[2]   Rahaf Aljundi[1]

## Abstract

Foundation models have transformed machine learning through large-scale pretraining and increased test-time compute. Despite surpassing human performance in several domains, these models remain fundamentally limited in continuous operation, experience accumulation, and personalization, capabilities that are central to adaptive intelligence. While continual learning research has long targeted these goals, its historical focus on in-weight learning, *i.e.*, updating a single model's parameters to absorb new knowledge, has rendered catastrophic forgetting a persistent challenge. **Our position is that combining the strengths of In-Weight Learning (IWL) and the newly emerged capabilities of In-Context Learning (ICL) through the design of modular memory is the missing piece for continual adaptation at scale.** We outline a conceptual framework for modular memory-centric architectures that leverage ICL for rapid adaptation and knowledge accumulation, and IWL for stable updates to model capabilities, charting a practical roadmap toward continually learning agents. *This work stems from discussions at the Dagstuhl Seminar on Continual Learning in the Foundation Model Era.*

[1]Toyota Motor Europe [2]University of Bremen [3]University of Florence [4]Microsoft Research [5]Università di Pisa [6]HEC Montreal, Mila–Quebec AI Institute, Canada CIFAR AI Chair [7]Bielefeld University [8]Georgia Institute of Technology [9]KU Leuven [10]University of Rochester [11]University of Texas at San Antonio [12]Nankai University [13]LUISS University [14]Stony Brook University [15]University of Tübingen [16]University of Trento, FBK [17]University of Groningen [18]Tsinghua University [19]Computer Vision Center Barcelona [20]Seoul National University. Correspondence to: Vaggelis Dorovatas <vdorovatas@hotmail.gr>, Rahaf Aljundi <rahaf.aljundi@gmail.com>.

*Proceedings of the 43rd International Conference on Machine Learning*, Seoul, South Korea. PMLR 306, 2026. Copyright 2026 by the author(s).

## 1. Introduction

Intelligence is often defined as the "ability to adapt to change" (Strauss, 2018), a perspective rooted in early work in psychology and neuroscience emphasizing adaptation, learning, and memory (Binet et al., 1905; Hebb, 1949), a view that remains echoed across a collection of modern definitions (Legg & Hutter, 2007). These principles underpin continual and lifelong learning, which study how agents adapt to new tasks while balancing stability and plasticity (Hadsell et al., 2020). Continual learning (CL) research has long studied the problem of accumulating knowledge over the lifespan of a model in an aim to bridge training- and test-time learning (Wang et al., 2024a; Verwimp et al., 2024), typically through parametric learning, referred to hereafter as In-Weight Learning (IWL), of a monolithic model. Despite the strong advancements, frequent parametric updates are challenged by forgetting (McCloskey & Cohen, 1989; French, 1999), optimization instability (Hadsell et al., 2020; Hess et al., 2024), and reduced plasticity (Dohare et al., 2024), rendering CL one of the most challenging machine learning problems.

In parallel, large-scale pretraining has dramatically extended model capabilities. While domain-specific adaptation was once viewed as essential (Wiggins & Tejani, 2022), foundation models now excel across a wide range of tasks and it becomes harder to identify entirely novel tasks. However, with the widespread adoption of AI agents (Luo et al., 2025) across a variety of scenarios, there is a growing need for such agents to operate continuously over extended periods of time. In doing so, they must accumulate knowledge and user interactions while maintaining an efficient computational footprint, underscoring the need for continual adaptation and accumulation of experience.

With advancing model capabilities, In-Context Learning (ICL) (Garg et al., 2022; Dong et al., 2024) emerged as an alternative learning mechanism to the traditional IWL paradigm. ICL modulates model outputs by incorporat-

---

*VD and MS are first and second authors; JC and MM are the last authors, with RA as the lead last author. The remaining authors are ordered alphabetically.

ing additional information, either raw inputs, retrieved or learned embeddings, via attention mechanisms (Vaswani et al., 2017). Recent studies suggest that ICL serves as a complementary learning mechanism (Lampinen et al., 2025; Dherin et al., 2025; Schuurmans, 2023; Russin et al., 2025), offering advantages over IWL in few shot generalization and potentially more effective incorporation of implicit patterns (Yin et al., 2024). Thus, most efforts on adapting LLM agents focus on extending context windows and build memory systems for storing interaction histories (Gao et al., 2026), typically assuming a frozen model. While this initially appears to overcome the need to defy forgetting, relying solely on ICL for adaptation faces significant limitations: over-reliance on large contexts leads to computational inefficiency with performance degradations as context length grows (Hong et al., 2025), while a frozen base model cannot adapt to fundamental shifts in data distributions or evolving user needs in non-stationary environments.

Here, we argue that the key to intelligent adaptation and knowledge accumulation lies in combining the strengths of the two learning mechanisms, ICL and IWL, under a modular memory architecture in which a pretrained core model is augmented with distinct memory modules: a working memory for active context and a long-term memory for rapid adaptation and knowledge accumulation. Rather than freezing the core model, long-term memory can be distilled through stable, low-frequency updates (counteracting forgetting), not to induce pure memorization in the core model but to enable higher-level generalization from accumulated knowledge and gradual performance improvement. **Our position is that adaptation has long been the missing cornerstone on the path toward intelligence. Now is the right time to unlock modular continual learning solutions that combine the complementary strengths of In-Context Learning and In-Weight Learning.** We outline the role of memory in related fields (Sec. 2), present our modular framework and its functionality (Sec. 3, Sec. 4) then call for action, discussing opportunities and application domains (Sec. 5).

## 2. Memory across Soft-, Hard- & Wetware

Memory is essential across various adjacent fields owing not only to its high relevance for consolidation of experiences, but also for adaptive compute. In the following, we briefly summarize complementary perspectives and highlight their relevance for continual learning agents at scale.

### 2.1. Early Use of Memory in Continual Learning

Memory has long been viewed as central to a machine learning system's ability to acquire new knowledge without erasing what was previously learned. Be-

fore the rise of deep learning, influential lifelong learning systems—such as the Never-Ending Language Learner (NELL) (Mitchell et al., 2018) and the Never-Ending Image Learner (NEIL) (Chen et al., 2013)—explicitly treated memory as a persistent repository accumulated over years of continuous operation, built from engineered components including web-scale extraction, coupled-pattern learning, morphological classification, and rule-based integration.

With the resurgence of neural networks, continual learning re-emerged around the challenge of catastrophic interference (McCloskey & Cohen, 1989; French, 1999). This shift led to a largely unified view of memory in deep continual learning, which can be broadly categorized into two forms: stored data and model parameters (De Lange et al., 2021; Wang et al., 2024a). First, rehearsal (or replay) of stored examples from past tasks became the dominant strategy for mitigating forgetting due to its empirical success (Rebuffi et al., 2017; Rolnick et al., 2019; Hayes et al., 2019; 2021; Verwimp et al., 2021; Wang et al., 2022a). Second, memory encoded in model parameters motivated approaches that constrain updates through parametric or functional regularization (Kirkpatrick et al., 2017; Aljundi et al., 2018; Li & Hoiem, 2017) or approximate replay via historical gradients (Lopez-Paz & Ranzato, 2017; Chaudhry et al., 2018; Aljundi et al., 2019b). Pseudo-rehearsal methods further synthesize past data using older model snapshots (Shin et al., 2017; van de Ven et al., 2020). Overall, memory in deep continual learning has been largely treated as a buffer with the primary role of mitigating forgetting during *model training*, rather than as an integral component of inference. While strong progress has been made, such sole reliance on In-Weight Learning is still largely challenged by forgetting and stability plasticity trade-offs.

### 2.2. Memory in Modern Large-scale Models

Current approaches to *external* memory in foundation models range from using the context window as working memory (Liu et al., 2022; Lewis et al., 2020) to agents with active memory management (Gao et al., 2026). They can be broadly classified into slot-based and distributed neural memories (see Sec. 4.1). Slot-based memories can be raw data or latent embeddings. Raw data can be viewed as libraries for future tasks, like human-interpretable code (Wang et al., 2023a; Zhang et al., 2025b) and text snippets (Suzgun et al., 2025b; Ouyang et al., 2025b). These libraries can be structured (Edge et al., 2024), curated (Suzgun et al., 2025b), generated by the model (Yu et al., 2025; Zhong et al., 2023), or exist as external knowledge bases (Lewis et al., 2020). Slot-based embedding memories can take the form of soft prompts (Zhang et al., 2025a), extracted activations (Wang et al., 2024b), compressed KV vectors (Chen et al., 2025), or learned KV vectors (Zhang et al., 2025a; Caccia et al., 2025; Eyuboglu

et al., 2025). Furthermore, a recent line of work implements slot-based memory structures inspired by human episodic memory (Fountas et al., 2025; Dong et al., 2025). In contrast, distributed neural memories are trained to generate contextual input to the main model (Behrouz et al., 2025b; He et al., 2024).

A common characteristic of these approaches is that adaptation is achieved almost exclusively through external memory and ICL, with the core model kept frozen. However, in slot-based methods this could lead to unbounded memory growth or aggressive compression that risks discarding critical information, while in neural distributed approaches, where memory is an external and continually updated neural module, this effectively shifts catastrophic forgetting to a different component of the system.

Another line of work is similar to traditional CL literature focusing on continually updating (parts of) the core model. This is done either via test-time training (Sun et al., 2024; Tandon et al., 2025) or by treating the core model as the system's single memory, targeting modularity and sparsity (Behrouz et al., 2025a; Lin et al., 2025a). However, they are again prone to catastrophic forgetting.

Finally, recent work (Liu et al., 2025) explores combining RAG with periodic fine-tuning, moving toward modular memory systems that jointly leverage ICL and IWL. However, its memory remains conversation-local and grows without principled mechanisms for consolidation, forgetting, or cross-conversation abstraction. Moreover, multi-modal information is reduced to textual summaries before parametric updates, effectively limiting learning to text-based representations and introducing information loss. Parametric updates rely on standard next-token prediction, which remains susceptible to memorization and forgetting (Chu et al., 2025; Shenfeld et al., 2026b), and evaluation is conducted in a static offline setup rather than a true continual learning setting with evolving distributions and long-term accumulation of experience. Overall, these limitations highlight that building scalable continual learning agents with principled modular memory fusion across diverse memory roles remains largely an open challenge.

### 2.3. Memory in Humans and Computers

**Human memory** is distributed across interacting systems operating at distinct timescales, capacities, and learning mechanisms, enabling rapid acquisition without catastrophic interference. The brain's memory stack demonstrates that continual learning can be both fast and stable, providing a possible frame of reference in favor of a *modular* architecture rather than a single store. Specifically, it consists of (i) *sensory memory* (very short-lived modality buffers), (ii) *working memory* and (iii) *long-term memory*, which is further split into *declarative* (concepts, facts,

events) and *nondeclarative* (skills, habits, priming) (Squire, 2004). A central organizing principle, alongside modularity, is *complementarity*: fast-learning systems such as the hippocampus encode pattern-separated, *multimodal* episodic representations, while slower cortical systems gradually acquire structured, generalizable knowledge in semantic long-term memory (McClelland et al., 1995). Coordination between these modules is mediated by *selective, temporally structured replay* (Hayes et al., 2021)—often during sleep—which supports consolidation across *multiple timescales* and enables stable learning with selective and graceful forgetting (Rasch & Born, 2013; Stickgold, 2005). Crucially, memory access is actively gated and different modules employ distinct representational formats [1].

**Computer architecture** treats memory as a first-class design constraint, where capacity is finite, access is non-uniform, and performance is dominated by *data movement* rather than compute. Modern systems mitigate finite resources by organizing memory into a hierarchy (registers, multi-level caches, DRAM, secondary storage) and by attaching explicit *policies* to movement across tiers: (i) **write policies** (*e.g.*, write-back to avoid unnecessary main-memory writes), (ii) **replacement/eviction policies** (*e.g.*, LRU/LFU approximations), (iii) **prefetching** (exploiting locality), and (iv) **telemetry** (misses, page faults) that makes resource exhaustion observable and actionable (Hennessy & Patterson, 2024). Critically, the success of these systems depends on *persistent memory with explicit management*. Memory reliability emerges because finite capacity is acknowledged and controlled, not implicitly overloaded. This view has recently been related to how continual adaptation places demands on memory organization and past experience reuse towards the design of continual accelerators (Kudithipudi et al., 2023).

**Implications for Continual Learning Agents.** Modularity, multiple timescales, and active memory management jointly define key desiderata. Drawing inspiration from human memory and computer architecture, several guidelines emerge: 1. Separate fast adaptation from slow integration: rapid, context-specific retrieval supports immediate behavior, while consolidation enables stable updates (Hayes et al., 2021). 2. Leverage multiple representational forms spanning episodes, abstractions, and skills (Squire, 2004; Tulving et al., 1972). 3. Differentiate declarative (facts, events) and non-declarative (skills) memory (Squire, 2004). 4. Treat memory as an actively managed resource: systems need policies for storage, replay, forgetting, consolidation, and contextualization (Dudai, 2004). Together, these elements remain underexplored in developing continual learning agents.

---

[1] We refer to the Appendix for an extended discussion of human memory structure and functions.

# 3. A Modular Memory Framework for Continual Learning Agents

Modular continual learning architectures have been conceptualized before (McClelland et al., 1995; Mitchell et al., 2018), but practical instantiations remained notoriously challenging. We posit that advances in foundation models now make their vision feasible. Specifically, the emergence of ICL as an effective learning mechanism (Yin et al., 2024; Lampinen et al., 2025) and the increasing ability of large models to reason with external knowledge (Luo et al., 2025) open the door to a new era of modular continual learning. Crucially, several recent positions have highlighted elements, such as episodic memory (Pink et al., 2025), metacognitive control (Liu & van der Schaar, 2025), and cognitive architectures for language agents (Sumers et al., 2023). We view these as components of our modular framework, unifying them into a coherent system for multimodal agents capable of continual learning.

## 3.1. Main System Components

We describe the different roles of the individual components in our proposed framework, as outlined in Figure 1.

**The core model** encodes general-purpose knowledge and skills that define the system's core capabilities. These capabilities are acquired during pretraining and are refined through consolidation after experience and knowledge accumulation. The core model can operate independently of long-term memory and provides fundamental capabilities such as perception, reasoning, multimodal understanding, action selection and generation, and tool use.

**The working memory** determines the current state of the system and serves to condition the main model on both the current environmental state and internal state. It may include both external signals, such as user instructions, demonstrations, or sensory inputs, and internal signals, such as retrieved long-term memories, intermediate reasoning traces, or latent planning states. Working memory is transient and limited in capacity.

**The long-term memory** stores information that persists beyond the current context, including facts, events, and personalized experiences (akin to declarative human memory, see 2.3). It is instrumental for rapid adaptation and accumulation of knowledge, reflection on captured experience and hence continual performance improvement. Long-term memory supports mechanisms for retrieval, updating, forgetting, and consolidation.

**The external world** represents the agent's own environment including accessible tools and other agents, which can be queried for auxiliary information and operations.

## 3.2. System Operation

The system operates in two regimes: an *external* interaction regime and an *internal* consolidation regime.

In the **external** regime (environment-driven), the system responds to signals originating from the environment. Incoming signals are encoded into the working memory, which conditions the core model. During inference, control mechanisms regulate the response process by implicitly determining a processing strategy, including whether retrieval from long-term memory is required and how much computational effort should be allocated to the current input (*e.g.* test-time scaling). When long-term memory is engaged, relevant past experiences or knowledge are retrieved into working memory to ground subsequent reasoning. Once the internal reasoning process completes, the system generates an appropriate response or action directed to the environment. The system also determines how this interaction (along with potential feedback from the environment) should be incorporated in the long-term memory.

In the **internal** regime (self-driven), the system focuses on consolidation and long-term memory refinement in the absence of external stimuli. During this phase, which occurs sparsely after accumulating sufficient experience, the model reprocesses and replays stored long-term memories to distill useful information into the core model, thereby refining existing capabilities and acquiring new skills. Consolidation improves efficiency by reducing reliance on explicit memory retrieval during future interactions. Concurrently, long-term memory itself undergoes refinement, compression and graceful forgetting, and the system reorganizes memories to better expose generalizable structures—simultaneously freeing capacity and enhancing the memory's ability to support future learning.

**Controlling the Flow.** As outlined, decisions about the system's operation require control policies. These may be specified via architect-designed heuristics or learned by the model. While intrinsic meta-cognitive learning is essential for autonomous continual learning systems, it remains an orthogonal research problem (Liu & van der Schaar, 2025). In Section 4.2 we discuss potential approaches and directions for both memory operations and control.

**Multiple Learning Mechanisms.** Our proposed memory framework supports learning with distinct mechanisms. Long-term and working memory enable fast learning through In-Context Learning. With ICL we do not explicitly refer here to using external raw data as additional input to the model, but rather to the attention mechanism that modulates the output of each layer based on additional embeddings capturing previous experiences, demonstrations, or task-specific information. In a round of interaction, relevant extracted information can be stored in

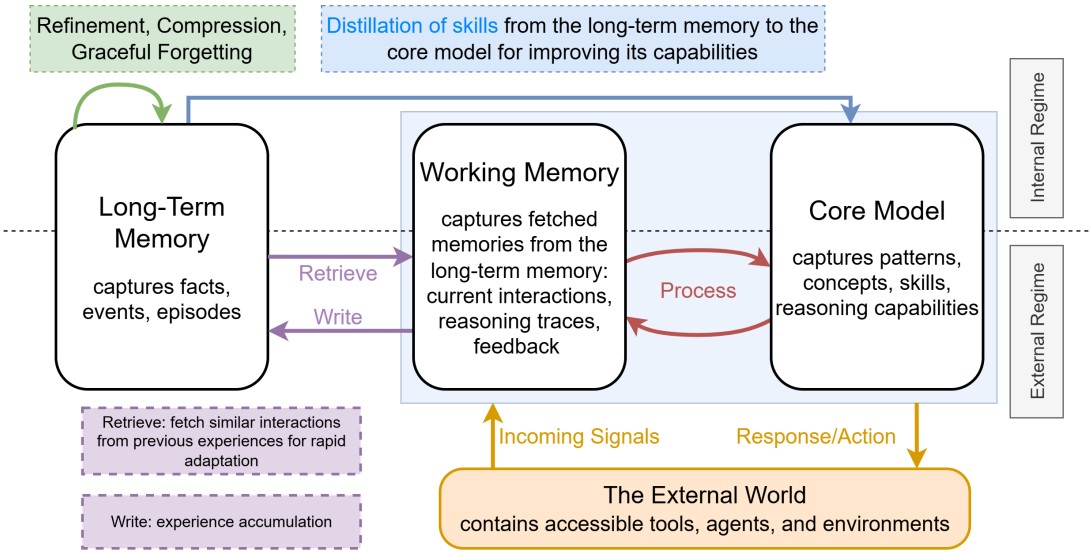

*Figure 1.* An illustration of the role of memory in unlocking continual learning through the construction of distinct modules. The framework comprises (1) a core model for perception and reasoning, (2) a working memory module for temporarily storing information relevant to the current interaction round, and (3) a long-term memory module that accumulates extracted experiences, facts, and observations. Knowledge stored in long-term memory is selectively retrieved into working memory and consolidated into the core model, steadily improving its capabilities.

the long-term memory, which can be retrieved in future interactions to immediately condition the core model, enabling few-shot adaptation and performance improvements without modifying the core model parameters. In contrast, the core model itself is updated more sparsely via In-Weight Learning, driven by consolidation during the internal regime. Consolidation distills information from long-term memory into the model's parameters, gradually improving its core capabilities and overall system's efficiency.

This multiple learning mechanisms view can also be paralleled with findings on memory consolidation and hippocampal replay in the brain, where experiences are rapidly encoded in the hippocampus and gradually consolidated into long-term cortical representations through replay dynamics (McClelland et al., 1995; Spens & Burgess, 2024). Analogously, our long-term memory coupled with ICL functions similarly to an episodic memory system, enabling rapid acquisition and retrieval of experiences, aligned with prior work linking ICL to human episodic memory (Ji-An et al., 2024), while the core model is updated gradually through replay-driven consolidation from long-term memory, drawing parallels with hippocampal replay [2].

# 4. Designing a Modular Memory

We now discuss how to design memory modules across two primary design dimensions: (1) memory representa-

tions, *i.e.,* how individual memory items are encoded, and (2) memory organization and functions, *i.e.,* how items are stored or retrieved and how memory is updated or consolidated. We focus on the rationale behind key choices and their implications with respect to desirable properties and design trade-offs. While no single design is universally fitting, we hope future research will target better memory structures and mechanisms. For a survey of existing memory design approaches, we refer to Hu et al. (2025).

## 4.1. Memory Representations

We start by describing the different memory representations and their attributes.

**(a) Slot-based memories.** They allocate discrete storage for each distinct memory (item[3]), ensuring independent accessibility. They consist of:

**(i) Raw Data.** Interactions, reasoning or input data would be stored in their original format. For example, text queries and responses for LLMs (Brown et al., 2020) or visual demonstrations with textual instructions for VLMs.

**(ii) Embeddings.** This format embeds raw data into a learned latent space, often compressing information or optimizing for specific functions like retrieval (Reimers & Gurevych, 2019). Three current main directions can be identified based on the representational space used: (1) *Activation-based* where hidden activations are extracted from the core model layer(s) when processing an input (Wang et al., 2024b); (2) *Key-Value caching*

---

[2]We refer to the Appendix for a discussion of additional connections to neuroscience.

[3] *E.g.,* a reasoning trace, a fact, or a tool description.

where attention layers' keys and values (Di et al., 2025; Chen et al., 2025) are used as memory representations, eliminating projection costs during retrieval (compared to (1)) but at the expense of doubled storage (key and value vectors must be stored); and (3) *Learned embeddings* where an additional learned transformation (*e.g.*, external encoders (Tack et al., 2024)) maps inputs into a specialized embedding space. Memory embeddings can also be optimized as learnable compressed vectors to the core model layers (*e.g.*, prefix tuning (Li & Liang, 2021), prompt tuning (Lester et al., 2021), slot attention (Locatello et al., 2020; Marconato et al., 2023)).

**(b) Distributed neural memories** encode information across shared network parameters, with no discrete boundaries between items. The defining characteristic is **cross-item interference**: multiple items modify overlapping weights, making it non-trivial to localize individual memories. Retrieval is performed implicitly through the network's forward pass.[4] An intuitive categorization of neural memory approaches is based on the model's update rule: 1) **Slow gradient-based updates in deep networks** perform updates iteratively using backpropagation over deep model layers (*e.g.*, Behrouz et al. (2025b)), while 2) **Fast associative updates** perform fast, often one-shot memory updates *e.g.*, Camelot (He et al., 2024), which implement extensions of the simple outer-product associative memory, storing memories outside the main model using associative rules. For a more detailed and theoretically grounded discussion we refer to Irie & Gershman (2026).

**Characteristics** that a memory structure should satisfy can now be defined, and the previously discussed representations can be evaluated against the desiderata in Table 1:

- **Per-Item Storage Efficiency** describes how compactly information is stored, ranging from verbatim storage in raw data to parameter-sharing in distributed memories.

- **Memory Capacity** refers to how storage requirements scale with the number of items stored. Slot-based methods have unbounded capacity that grows with the number of items, whereas distributed memories' capacity is typically bounded by the network size and architecture.

- **Modality** indicates whether individual item representations are unimodal (*e.g.*, a single image or text) or can encode multimodal information. Raw data are naturally constrained to unimodal item representations.

- **Update Speed** measures how quickly new items can be robustly incorporated into memory. Slot-based methods support fast updates via direct slot addition. For neural

methods, update speed depends on the update rule.

- **Selective Forgetting** indicates the ease of removing specific items from memory without interference. Slot-based methods enable easy instance-level deletion by removing individual slots. Distributed memories hinder selective forgetting because entangled shared parameters make removing one item risk harming others.

- **Retrieval** characterizes the cost of finding items given a query. In slot-based memories, retrieval cost grows with the number of stored items and depends on the underlying data structure (*e.g.*, linear for lists, logarithmic for trees), whereas in distributed memories it is constant with respect to item count and depends on network size.

- **Model transferability** refers to whether memory representations can be reused across agents with different core models (*e.g.*, experience sharing in multi-agent systems). This is easy for raw data (modulo privacy constraints) but challenging for model-specific representations: for instance, KV caches are not directly interpretable by other agents, although recent work shows sharing is possible among agents with the same LLM (Ye et al., 2025). For neural memories, model merging or alignment techniques provide potential mechanisms (Yang et al., 2024).

- **Generalization** is a key property of memory, referring to its ability to support reasoning across related scenarios by enabling efficient retrieval and updating of relevant past experiences for new inputs. Raw data memories offer limited generalization, whereas learned and neural representations can abstract and reuse experience more effectively. For instance, Chen et al. (2025) show that storing reasoning traces as KV caches yields better experience reuse on related problems than context summarization.

**Discussion.** Memory representations trade off fidelity, efficiency, and generalization. Raw data preserves complete information and is model agnostic, but requires high storage, incurs costly retrieval that often necessitates embedding for similarity based search, and does not directly generalize across scenarios. Latent embeddings compress information for efficient similarity based access and encode relational structure that supports generalization, but are lossy and architecture specific. Neural memories provide high storage efficiency, efficient retrieval, and strong generalization, but suffer from interference leading to forgetting along with poor support for selective removal.

## 4.2. Memory Structure and Functions

Memory organization ranges from flat lists, which represent memories as temporal sequences (Lewis et al., 2020), to hierarchical structures that organize memories into layers of abstraction such as trees (Sarthi et al., 2024), and graph-based structures that explicitly

---

[4]While methods like prefix tuning optimize embeddings directly, we categorize them as slot based rather than neural since each item's parameters are independent, differently from grouping all parameter-based storage together as in Hu et al. (2025).

| Memory Type | Per-item Storage | Capacity | Modality | Update Speed | Interference Risk | Selective Forgetting | Retrieval | Generalization |
|---|---|---|---|---|---|---|---|---|
| **Slot-based Memory** | | | | | | | | |
| ↪ **Raw Data** (summaries, exemplars, coresets) | Low | Unbounded | Uni-modal | Fast | Low | Easy | Grows with #items | Limited |
| ↪ **Embeddings** (KV cache, activations, learned) | Medium | Unbounded | Uni/ Multi-modal | Fast | Low | Easy | Grows with #items | Implicit (learned encodings) |
| **Neural Memory** (*e.g.*, DNNs, Hopfield, SDM) | High | Bounded | Uni/ Multi-modal | Slow/ Fast | High | Difficult | Depends on network size | Explicit (shared params) |

*Table 1.* **Desiderata and assessment of current memory representations (as defined in Sec. 4.1).** Storage is more compact for neural memory while Capacity is unbounded for slot-based, raw data is typically unimodal, update speed is varied based on specific design choices, interference is high in neural memory, selective forgetting is easier for slot based. Finally generalization is poor in raw-data, implicitly encoded in embeddings, while explicitly optimized in neural memory.

encode relations among items, enabling richer semantic access (Jimenez Gutierrez et al., 2024). Memory should be structured to support efficient and effective retrieval, updates, and selective forgetting. Our aim is not to advocate for a specific structure, but to articulate desired functional properties that can guide future research.

**When and What to Add.** Memory updates operate across **multiple timescales** in our modular memory system. Working memory is highly transient, retaining information only within a task or context, while long-term memory evolves more slowly as information is transferred from working memory via capacity-based or event-driven updates. At the slowest timescale, the base model is updated only after substantial experience accumulation during consolidation phases, yielding less frequent updates thereby naturally reducing exposure to interference. Update decisions can **cascade across memory modules**: committing items to long-term memory clears working memory, while absorbing long-term memories into model parameters enables episodic traces to be compressed or eventually evicted. Update policies may be governed by **designer-specified heuristics** (*e.g.*, capacity thresholds, periodic consolidation & event-driven signals (Suzgun et al., 2025a)) or by **learned policies** (Nawrot et al., 2024). Key challenges include objective design, long-horizon credit assignment, and policy adaptation. Determining what to store in neural memory has been extensively studied in continual learning (*e.g.* coresets (Blömer et al., 2016; Har-Peled & Kushal, 2005) and heuristic selection (Mirzasoleiman et al., 2020; Killamsetty et al., 2021; Paul et al., 2025)), and recently in LLMs (*e.g.*, context summarization (Yu et al., 2025; Anthropic, 2025) and KV-cache selection (Devoto et al., 2024; Zhang et al., 2023), yet the question remains open.

**When and What to Forget.** These questions are long-standing challenges in the CL literature related to limited model capacity and constrained computational resources (Aljundi et al., 2018). Forgetting occurs either when capacity limits are reached or according to periodic schedules. It can take the form of **eviction** or **compression** (*e.g.*, merging, summarization). Beyond constraints, forgetting may be also **deliberately induced** to address outdated, conflicting, or ethically sensitive information (*e.g.*, Hegde et al. (2025); Qi et al. (2023); Guan et al. (2025)). **Temporal policies** (*e.g.*, FIFO queues) (Lopez-Paz & Ranzato, 2017) evict or compress items based solely on recency, offering simplicity but risking the loss of valuable long-term information, whereas **importance-based heuristics** (Zhang et al., 2023; Dorovatas et al., 2025) score items using signals such as retrieval frequency or cumulative attention. In contrast, **learned policies** (Bui et al., 2025; Łańcucki et al., 2025) optimize forgetting by predicting future utility under memory constraints, for example via objectives that jointly penalize task loss and memory size, encouraging sparse selective retention.

**When and How to Retrieve.** The simplest strategy retrieves on every input, maximizing information availability while incurring retrieval overhead and potential noise injection. More selective approaches trigger retrieval based on internal model signals such as uncertainty (Chen et al., 2026), or via learned retrieval decisions in which the model explicitly generates a retrieval request (Wang et al., 2023b). In contrast, *how* to retrieve is constrained by the memory representation. In neural memories, retrieval is implicit and learned as part of the forward pass, whereas slot-based memories require explicit retrieval mechanisms. **Similarity-based** retrieval is the dominant paradigm (Lewis et al., 2020), including online and rehearsal-based methods (*e.g.*, (Aljundi et al., 2019a; Shim et al., 2021)). On the other hand, **frequency-based** retrieval maintains access counts and prioritizes frequently used items (Lin et al., 2025c).

### 4.3. Core Model Consolidation

Consolidation aims to update the pretrained model, already equipped with strong perception and reasoning capabilities, by leveraging experience accumulated in the long-term

memory. Rather than promoting shallow memorization, such consolidation is envisioned to refine the model's capabilities and internalize new skills, enabling higher-level generalization, adaptation to distribution shifts, and improved overall performance. Updates to the core model rely on IWL. Thus, effective consolidation depends on appropriate IWL policies over stored experiences for which continual learning approaches such as regularization and replay can be adapted. Recent work has investigated how new knowledge is acquired and consolidated (Dohare et al., 2024; Shah et al., 2025), proposing training recipes that promote generalization (Eyuboglu et al., 2025; Padmanabhan et al., 2023) and analyzing the generalization/memorization trade-offs (Chu et al., 2025).

We believe robust consolidation lies at the intersection of *three complementary ingredients*. First, memory representations should store compressed encodings of experiences rather than raw trajectories, enabling built-in generalization and more efficient consolidation. Second, architectures and consolidation mechanisms should promote parameter stability, specialization, and modularity so that updates remain localized to relevant subnetworks (Shazeer et al., 2017; Aljundi et al., 2018; Panos et al., 2025; Dorovatas et al., 2026). Third, training objectives should explicitly encourage generalization while reducing forgetting. Recent findings have demonstrated that on-policy methods (e.g. on policy distillation (Shenfeld et al., 2026a; Hübotter et al., 2026) or on-policy reinforcement learning objectives) forget less and generalize better than supervised fine-tuning (Chu et al., 2025; Shenfeld et al., 2026b). Together, these ingredients point toward consolidation regimes that refine capabilities while preserving previously acquired knowledge.

To address the question of *when consolidation should be triggered*, practical systems must balance optimization stability with computational efficiency. Here, the human brain offers a useful reference point: consolidation emerges from temporally structured replay during sleep, coordinating fast and slow learning systems. Early work on LLMs has already begun exploring analogous ideas using offline or inactive periods for replay and adaptation (Lin et al., 2025b), closely aligning with our envisioned internal consolidation regime. Such mechanisms may provide a natural way to periodically integrate accumulated experience into the core model while minimizing interference with ongoing adaptation.

Overall, since parametric consolidation into the core model occurs fairly infrequently and with the aid of a long-term memory and proper consolidation mechanisms, the risks associated with continual IWL, e.g., forgetting, optimization instability and loss of plasticity, are naturally reduced.

## 5. Challenges and Opportunities

We outline a call to action to address emerging opportunities and challenges across key application domains.

### 5.1. Opportunities for Memory-Centric Agents

- **Faster Adaptation & Efficiency.** Maintaining long-term memories facilitates efficient learning of new knowledge by leveraging similarities between novel and past experiences. Drawing on previously encountered, related scenarios enables faster problem solving and reduces the need for explicit deliberation (Wu et al., 2025).

- **Hallucination Reduction & Alignment.** Explicit memory supports grounded responses by requiring models to reference factual memories as a form of "certificate of understanding," thereby reducing hallucinations (Kollias et al., 2024). When hallucinations occur, user corrections can be stored to calibrate the model's future outputs.

- **Explainability.** Requiring models to reference memories when producing outputs improves explainability *e.g.*, by enabling explicit descriptions of prototypical cases (Rymarczyk et al., 2023). These grounded memories help explain model successes and failures, clarifying *why* a particular output was produced.

- **Personalization.** Memory management and retrieval enables personalization (Chen et al., 2024) of assistive agents to users' needs and preferences. A user's requirements vary over time, and graceful forgetting can enable access to the most up-to-date knowledge.

- **World Modeling & Future Prediction.** Explicit long-term memory complements world modeling, viewed as a milestone in embodied intelligence that enables agents to predict future states and act accordingly (Ha & Schmidhuber, 2018; LeCun et al., 2022; Sekar et al., 2020), by grounding predictions in an agent's experience. Here, deviations between predicted and observed outcomes provide a learning signal and memory can capture agent-specific dynamics shaped by past interactions, rather than approximating the full environment distribution.

### 5.2. Challenging Application Domains

In **Continual Learning of Tasks, Test-Time Adaptation**, models face domain shifts and are updated online after deployment, making optimal parameters context-dependent (Wang et al., 2022b). However, existing methods (Liang et al., 2025) rely on data augmentation, self-supervised objectives, or stochastic weight resets, leaving contextual and domain-aware memory management largely unexplored.

In **Open-Ended and Open-World Learning**, models encounter new tasks (*e.g.*, new concepts in the context of classification) both at train and test time (Bendale & Boult,

2015; Mundt et al., 2023). Domain-aware memory with ICL makes it natural to transcend predefined label spaces.

In **Tool Learning**, agents benefit from accumulating and reusing knowledge about tool interfaces, compositions, and effective usage over time. Whereas static models can orchestrate tools (Shen et al., 2023; Patil et al., 2023), augmenting them with memory enables continual improvement in tool usage as the ecosystem evolves. Rather than relearning affordances, memory allows agents to store successful tool-use traces and learn what to offload externally.

In **Capturing of Reasoning and Experience**, test-time scaling (Snell et al., 2025) has shown strong gains on challenging tasks by increasing test-time compute and enabling models to generate longer reasoning traces. However, these traces are typically discarded after each interaction, forcing repeated recomputation for similar reasoning. A CL agent equipped with modular memory can reuse past reasoning traces, enabling increased efficiency in inference and performance improvement as experience accumulates over time (Ouyang et al., 2025a; Chen et al., 2025).

In **Streaming Scenarios**, the sequential, long-horizon, and resource-constrained nature poses canonical CL challenges, *e.g.,* video understanding. Recent work increasingly emphasizes explicit memory mechanisms to retain, retrieve, and reason over long temporal contexts in online settings (Kang et al., 2024; Xiong et al., 2025; Dorovatas et al., 2025; Xiong et al., 2025), highlighting the relevance of our modular memory framework for these challenges.

In **Embodied Agents**, monolithic neural networks often lead to struggles with long-horizon action rollouts, as they must retain knowledge of past observations and actions to generate contextually appropriate behavior. Scene graphs have been proposed as pre-established memories that capture object relations under full observability (Mohammadi et al., 2025; Honerkamp et al., 2024). However, real-world environments are observed incrementally and evolve over time. Recent work therefore treats learnable memory as a task objective (Gupta et al., 2025). A continual-learning agent with explicit memory can recall object relations for planning and update them as the environment changes.

## 6. Alternative Views

**Infinite Context & Long Context + RAG.** An alternative view might argue that extending context windows to include past information is sufficient, either by architectural advances or retrieval-augmented generation (RAG) (Lewis et al., 2020). While the conceptual ease may be appealing, this paradigm faces several limitations. First, processing extended contexts remains computationally expensive, with inference costs increasing as context length grows, despite recent advances in long-context architectures (Co-

manici et al., 2025). Additionally, long-context reasoning performance degrades as context grows and can unpredictably alter model behavior (Liu et al., 2024; Du et al., 2025; Geng et al., 2025). Additionally, RAG is sensitive to surface-level linguistic variation (Cao et al., 2025) and is susceptible to noise due to its static nature and the absence of mechanisms for selective forgetting. Finally, storage requirements are exacerbated in multi-modal and embodied settings, where storing and retrieving raw sensory experiences is prohibitively expensive.

**Continual Parametric Updates.** Another perspective could posit that continually updating the core model, which lies at the heart of traditional CL approaches ((section 2.1)) and underpins the recent test-time training paradigm (Sun et al., 2020), is sufficient to absorb new knowledge. Whereas efficient techniques exist for sparse updates (Han et al., 2024), this strategy is costly at scale, requires careful per-update hyperparameter tuning, and is prone to catastrophic forgetting. Even when using isolated modules such as LoRA adapters (Hu et al., 2022), merging updates remains challenging, and low-rank updates can exacerbate forgetting (Shuttleworth et al., 2024). As emphasized throughout this paper, updating models through IWL remains very challenging, which makes solely relying on continual parametric updates problematic.

From a **practical cost perspective**, we believe that combining ICL and IWL offers a more efficient trade-off over long-term operation than either mechanism alone. Long-context inference is memory- and compute-intensive, whereas continual full-model updates are costly and unstable; their combination enables ICL to support rapid adaptation with reduced need for frequent parameter updates, while IWL periodically consolidates knowledge to control context growth and amortize retrieval overhead over time.

## 7. Conclusion

We argue that the future of adaptive AI needs to integrate the strengths of two learning paradigms via modular memory: (a) in-context learning through complementary working memory and long-term memory, and (b) in-weight learning through low-frequency parametric updates. Rather than choosing between them, we posit that a smooth knowledge transition pathway is necessary: experience first enters working memory for immediate adaptation, accumulates in long-term memory for selective retrieval and refinement, and is gradually consolidated into the model parameters. We ground our framework in functional considerations for memory modules. As memory-centric and continuously updated models develop somewhat independently of established continual learning research, we call for realignment and assessment of their strengths, weaknesses, and synergies on the path toward continual learning agents.

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

# A. Human Memory & Neuroscience

## A.1. Human Memory

The brain demonstrates that continual learning can be both fast and stable, but not from a single memory store. Instead, memory is split across systems with different time scales, capacity limits, and learning rules (Squire, 2004; Atkinson & Shiffrin, 1968). For broader background on memory systems, working memory, and consolidation, see references (Squire, 2004; Baddeley, 2000; Dudai, 2004; Rasch & Born, 2013; Diekelmann & Born, 2010). A basic taxonomy from cognitive psychology literature distinguishes:

**A basic taxonomy.** Cognitive psychologists distinguish among (i) *sensory memory* (very short-lived modality buffers), (ii) *working memory* (a small-capacity system for holding and manipulating task-relevant information), and (iii) *long-term memory* (information retained across hours to years) (Atkinson & Shiffrin, 1968; Baddeley, 2000). Long-term memory is often split into *declarative* (explicit) and *non-declarative* (implicit) systems (Squire, 2004). Declarative memory supports learning and recalling concepts, facts, and events; non-declarative memory supports the learning and execution of skills, habits, priming, and conditioning (Squire, 2004).

**Working memory and control.** Working memory is closely tied to *cognitive control*, the processes that allocate attention, suppress distractors, and decide when to retrieve stored information. In the Baddeley (2000) model, a *central executive* coordinates specialized buffers and an *episodic buffer* that binds information across sources and links to long-term memory. For AI systems, the main point is that memory use is *gated*: only some inputs are stored, and retrieval is an active choice.

**Declarative memory: episodic, semantic, autobiographical.** Declarative memory is often divided into *episodic memory* (events with time/place context) and *semantic memory* (facts, concepts, and general knowledge) (Tulving et al., 1972; Squire, 2004). These interact: repeated or important episodes can become semantic knowledge, and semantic structure shapes how episodes are encoded and generalized. *Autobiographical memory* combines episodic details with stable personal facts and self-relevant knowledge; it supports long-run consistency of goals and identity (Conway & Pleydell-Pearce, 2000).

**Complementary learning systems and replay.** Complementary learning systems (CLS) theory explains how the brain balances plasticity and stability (McClelland et al., 1995). In CLS, the hippocampus learns quickly from new experience by caching latent episodes, while the neocortex is slowly updated from the replay of these latent episodes to build structured knowledge. A key idea is *representational complementarity*: hippocampal codes are more *pattern-separated* (sparse, low overlap) to reduce interference, while cortical codes are more *distributed* (overlapping) to support generalization. Consolidation is often linked to sleep and involves *reactivation* and *replay* (reinstating prior activity patterns), which can drive systems-level integration (Rasch & Born, 2013; Diekelmann & Born, 2010; Stickgold, 2005). Consolidation spans *synaptic consolidation* (minutes–hours) and *systems consolidation* (days–years) (McGaugh, 2000; Dudai, 2004). Replay is a mechanistic link between episodic storage, retrieval, and long-run integration, and it is not just i.i.d. rehearsal; it can be selective and temporally structured (Hayes et al., 2021). Unlike replay implementations in continual learning, the hippocampus stores multi-modal latent temporally correlated sequences (episodes) which are replayed to transfer knowledge into the neocortex to form generalizable long-term concepts.

## A.2. Connections to Neuroscience and Biological Plasticity

As discussed, our proposed framework is broadly inspired by the separation of memory systems and consolidation processes observed in biological intelligence. Several recent directions in neuroscience and biologically inspired machine learning provide additional useful conceptual parallels and motivating principles for the mechanisms discussed throughout this work:

**Short-term synaptic plasticity and working memory.** Short-term synaptic plasticity (Miconi et al., 2018) and fast Hebbian adaptation (Ba et al., 2016) are especially relevant to the working-memory and in-context learning (ICL) components of our framework. In these settings, functional connectivity changes rapidly as a function of recent neural activity without requiring permanent modification of long-term parameters. Conceptually, this aligns closely with Fast Weight Programmers (FWPs) (Irie & Gershman, 2026), which separate slow stable parameters from fast input-dependent connections. Modern attention mechanisms can be interpreted as implementing a restricted form of such fast plasticity, where activations dynamically modulate effective connectivity during inference. However, in current transformer architectures this adaptive modulation is largely confined to attention layers, while feed-forward modules remain effectively non-plastic. From this perspective, biologically inspired short-term synaptic plasticity mechanisms may offer a promising direction for developing richer and more flexible working-memory systems that extend beyond standard attention-based architectures.

**Behavioral-timescale plasticity and continual learning.** Another relevant line of work concerns behavioral-timescale synaptic plasticity (Magee, 2026), which studies how credit assignment and synaptic updates evolve over timescales associated with ongoing behavior and adaptation. At a high level, this connects to a broader research direction focused on rethinking the learning algorithm itself for continual learning settings. In the continual learning literature, methods such as Continual Backprop (Dohare et al., 2021) modify the update rule to mitigate loss of plasticity and improve long-term adaptability. We view biologically inspired learning algorithms of this kind as complementary to architectural and memory-based approaches, and as a promising source of inspiration for future work on core-model consolidation and continual adaptation in foundation models (as discussed in the main manuscript, learning algorithm is one of the three core ingredients for effective consolidation).

**Low-rank structure, modularity, and specialization.** Recent work on low-rank recurrent neural networks (RNNs) (Driscoll et al., 2024) is also highly relevant to our discussion of consolidation and architectural modularity. These studies suggest that low-rank or structured connectivity constraints can promote task specialization, compositionality, and improved generalization within neural systems. Such findings directly support our argument that effective consolidation mechanisms should encourage parameter specialization and modularity, thereby localizing updates to relevant subnetworks and reducing interference between skills. More broadly, these results provide a neuroscience-grounded motivation for architectures that combine stable shared representations with specialized adaptive submodules, which may ultimately help mitigate forgetting during long-term continual learning and consolidation. As discussed in the main paper, architecture and representations are another crucial ingredient shaping consolidation.

