# OpenReview forum: "Position: Modular Memory is the Key to Continual Learning Agents"
_ICML.cc/2026/Position_Paper_Track — ICML 2026 Position Paper Track spotlight_

### Official Review · Reviewer_rg3q · 2026-03-12

**Significance:** 3
**Argument Clarity:** 3
**Rating:** 4
**Confidence:** 4

**Questions:**

1. Please see to Weakness #2 for first question.

2. The proposed framework combines ICL-based memory retrieval with sparse IWL-based consolidation into the core model. In practice, a similar setup could be achieved by combining RAG (for retrieval/ICL) with periodic fine-tuning on accumulated interactions (for IWL). Such systems already exist (e.g. sth similar to [1]). What does the proposed modular memory framework offer over this simpler baseline, and what would be concretely lost by using the simpler approach instead?

 [1] MemVerse: Multimodal Memory for Lifelong Learning Agents, Liu et al., 2025

**Alternative Views Section:**

Yes

**Compliance With Llm Reviewing Policy A Conservative:**

Affirmed.

**Discussion Potential:**

3

**Final Justification:**

My questions were partially addressed in the rebuttal.

Therefore, I am leaning towards accept.

**Paper Summary:**

This position paper argues that neither pure In-Weight Learning (IWL) nor pure In-Context Learning (ICL) alone is sufficient for building truly adaptive continual learning agents, and that modular memory, combining a working memory module for rapid ICL-based adaptation with a long-term memory module for knowledge accumulation, alongside sparse IWL-based consolidation into a core model, is the missing architectural ingredient.

The paper grounds its position in analogies to human memory systems and computer architecture memory hierarchies, surveys existing memory representations (slot-based vs. distributed neural), and concludes with a call to action across several application domains including embodied agents, streaming scenarios, and tool learning.

 A notable aspect outlined by the article is the proposed smooth knowledge transition pathway: experience enters working memory, accumulates in long-term memory, and is eventually consolidated into model parameters at low frequency. The authors aim to bridge the currently (somewhat) disconnected research communities of continual learning and memory-augmented foundation models.

**Position:**

Yes

**Position In Title:**

Yes

**Related Work:**

3

**Strengths And Weaknesses:**

Strengths:

1.	The paper shows a broad coverage of related topics, spanning classical continual learning, computer architecture, neuroscience, and modern LLM agent research. The connections between fields are also substantive and not superficial.

2.	The argument on that so far the field has (at least implicitly) treated ICL and IWL as competing paradigms and not complementary ones is a key issue. Especially since now AI agents are moving into real-world deployment, limitation of both frozen-model ICL and IWL are becoming more visible.

3.	Both alternatives discussed (infinite context/RAG and continual parametric updates) are positions that are actively hold in the community and the rebuttals they make in this section is well-grounded in the literature

Weaknesses:

1.	The main issue is one of the most important components proposed in the position (the consolidation mechanisms) remain quite under-developed. Section 4.2 explains this at high-level and provides generic insight on how this would be achieved but doesn’t discuss concreted objectives, update triggers, or safeguards (for e.g. catastrophic forgetting) for this critical component. While the authors also mention this is an open research direction, more specificity would have strengthened the contribution.

2.	Modular continual learning architectures have been conceptualized before (some works include the ones cited and acknowledged by the authors) in the opening paragraph of Section 3), and recent work has highlighted episodic memory, metacognitive control and cognitive architectures as components of similar frameworks. The contribution here mainly focuses on unifying them into a coherent system. This is valuable, but it remains unclear what is concretely new beyond this unification. Some further clarification, for example clarifying “what a researcher would do differently by following this framework, compared to the cited prior work?”, would strengthen the novelty.

**Support:**

3

---

> ### Author Rebuttal · Authors · 2026-03-31
>
> We thank the reviewer for their constructive feedback and positive assessment.
>
> ---
> **Consolidation**
> ---
>
> We agree with the reviewer that consolidation remains critical and challenging, and we highlight what we already discuss in Section 4. **Our key argument is that by utilizing ICL for short-term adaptation, the need for frequent parametric updates is significantly reduced, naturally alleviating many interference and instability challenges if approached carefully**. As discussed in Section 4, **consolidation is envisioned to refine the model's capabilities and internalize new skills promoting higher-level generalization rather than shallow memorization** through appropriate learning policies, training recipes, and the natural adoption of CL mechanisms such as regularization and replay to preserve prior knowledge.
>
> To elaborate, we believe robust consolidation depends on three promising ingredients:
>
> - **Memory representations** should store compressed encodings of experiences rather than raw data, providing built in generalization and supporting consolidation.
> - **Architectures and consolidation mechanisms** should promote parameter stability, specialization, and modularity so that updates are localized to relevant subnetworks [1,2].
> - **Training objectives should promote generalization and reduce forgetting**. Beyond distillation methods, e.g., [3], recent work shows that on policy RL objectives forget less and generalize better than SFT [4,5].
>
> Regarding **consolidation triggers**, we outline possible directions in Section 4 and believe **they should balance optimization stability and computational efficiency**. As noted in Section 2.3, the human brain offers a useful reference: consolidation is driven by temporally structured replay during sleep, coordinating fast and slow learning. Early LLM work already explores similar ideas using offline/inactive time [6], aligning with our internal consolidation regime.
>
> We will integrate a more explicit discussion of these points into the revised manuscript,
>
> ---
> **Answering Questions**
> ---
> **Q1**: While related works highlight episodic memory, metacognitive control, and cognitive architectures as individual components, we explicitly unify these under the lens of Continual Learning at scale, bridging traditional CL research and modern memory-augmented foundation models,  two communities that tend to overlook each other's challenges, as we emphasize in our conclusion. A researcher following this framework would reason differently in several concrete ways:
> - **Design solutions that integrate both learning mechanisms**, not just add memory to a frozen model, using ICL for rapid adaptation and IWL for slower consolidation (Sections 3 and 4).
> - **Develop novel memory structures** tailored to challenging domains beyond text, as discussed in Section 5.
> - **Define explicit policies** e.g., for what to store, what to retrieve, and when to update long term memory, rather than relying on simple accumulation (Section 4).
> - **Build robust consolidation mechanisms** that improve core model capabilities from stored memory **without drifting into pure memorization** (Section 4).
>
> **Q2**:  **On works combining RAG and Periodic Fine tuning (e.g., MemVerse)**. We appreciate the reviewer highlighting MemVerse as a relevant recent work. It is indeed a step toward acknowledging both ICL and IWL and building modular memory systems. However, viewing it through our framework makes several limitations clear:
> - The long term memory creates a new graph for each conversation without principled policies for updating, forgetting, or relating experiences across conversations, reflecting the unbounded growth issue discussed in Section 4.
> - Consolidation is text-only: multimodal information is compressed into captions and transcriptions before any parametric update, meaning the system is effectively ICL-only in the multimodal domain, with text compression introducing inevitable information loss.
> - The parametric updates rely on simple next-token prediction, which as we discuss in Section 4 are prone to memorization rather than generalization, with no mechanism to shield against forgetting.
>
> Importantly, MemVerse is evaluated **in an offline, static setup** rather than a true CL scenario, remaining within the memory augmented architecture frontier with ad hoc updates and without addressing forgetting, distribution shift, or long term accumulation of experience. Nevertheless, MemVerse is a valuable step toward recognizing the ICL–IWL duality and the need for modular memory. Our framework is intended as a reference point for evaluating and improving such approaches, not excluding them, and we hope it helps the community diagnose limitations and guide future progress.
>
> ---
> [1] https://arxiv.org/abs/1701.06538
>
> [2] https://arxiv.org/abs/1711.09601
>
> [3] https://arxiv.org/abs/2601.19897
>
> [4] https://arxiv.org/abs/2501.17161
>
> [5] https://arxiv.org/abs/2509.04259
>
> [6] https://arxiv.org/abs/2504.13171

---

> > ### Author Rebuttal · Reviewer_rg3q · 2026-04-01
> >
> > My concerns have been adequately addressed.

---

### Official Review · Reviewer_FSdX · 2026-03-13

**Significance:** 4
**Argument Clarity:** 3
**Rating:** 6
**Confidence:** 4

**Questions:**

- I would have imagined works on integrating short-term synaptic plasticity into neural networks would have been very appropriate to discuss here. One paper that comes to mind is this: https://proceedings.mlr.press/v80/miconi18a.html?ref=https://githubhelp.com. In that sense, people have been thinking about continual adaptation in neural networks; but I did not see this aspect being discussed. Could authors clarify if these lines of works are not consistent with what they are proposing? If they are consistent, a more explicitly acknowledgement (and proper citations to tens of works in this line) is needed that people have been thinking about this concept.

- Second, neuroscience literature has been thinking about this problem a long time now in the context of synaptic plasticity. In fact, a recent hot topic is the behavioral time-scale plasticity, which seems conceptually very relevant here: https://www.nature.com/articles/s41593-026-02214-2. can you also make connections to this line of research?

- Yet another line of research seems to be the work by computational neuroscientists on low-rank RNNs, where they have found rank-based constraints (unlike LORA, here ranks are assigned to distinct tasks as opposed to post-trained) could enable generalization. An important anchor could be this: https://www.nature.com/articles/s41593-024-01668-6

**Alternative Views Section:**

Yes

**Compliance With Llm Reviewing Policy A Conservative:**

Affirmed.

**Discussion Potential:**

3

**Final Justification:**

As noted in my review, I comfortable recommending a strong accept. This position paper was the strongest in my batch, caters to two main communities, and presents new ideas on both of these hot fields.

**Paper Summary:**

Traditional methods for pre-training LLMs rarely shows signs of adaptability that we associate with intelligence. There are methods for continual learning, but more often than not they lead to catastrophic forgetting. To make progress in this direction, this position paper calls for a marriage between in-weight learning and in-context learning paradigms. They propose a framework around three key components, which interact with each other and have their own learning processes. The memory components "learn" in the sence of ICL, whereas the cor model learns in the sense of IWL. This idea seems somewhat reminiscent of memory consolidation in the brain and hippocampal replay, though not particularly presented as such.

**Position:**

Yes

**Position In Title:**

Yes

**Related Work:**

2

**Strengths And Weaknesses:**

Strengths:

- I believe the topic is somewhat ambitious and as such quite interesting to the broader ML community, and potentially to outside researchers from neuroscience and neurobiology. Continual learning is a core topic of interest in many fields that has remained elusive for a long while.

- The proposed position is clearly articulated, the evidence for its relevance *and* directions for how to move forward are clearly presented.

- There are clear connections to neuroscience, specifically how neuroscientists think about certain biological processes (see summary) in the brain. These connections make this position paper broadly appealing, but also provides circumstantial support/hope that what is proposed could be a viable solution.

- Section 5 in itsel is a gem. There are several non-trivial connections, and by itself, that section deserves publication imho. For instance: "Requiring models to reference memories when producing outputs improves explainability" and "Explicit memory supports grounded responses by requiring models to reference factual memories".

Weaknesses:

- The only real weakness I find is that several works on short-term synaptic plasticity are ignored in the presentation (see my question below).

Overall assessment:

While I have my own opinions on the position the authors have presented, authors presented their ideas clearly enough that I can form these opinions. This is a hallmark of a good position paper.

**Support:**

4

---

> ### Author Rebuttal · Authors · 2026-03-31
>
> We sincerely thank the reviewer for these valuable references and for the positive assessment. Connecting our framework to neuroscience and computational neuroscience is an important point that we wanted to highlight. Due to space constraints in the submission, we limited the neuroscience discussion (human memory in Section 3.2) to recent evidence that directly aligns with our modular memory framework and serves as inspiration for continual learning agents at scale, given that humans remain the strongest empirical example of continuous adaptive intelligence. We very much welcome additional references that the reviewer believes would relate to the framework or mechanisms of each component. Below, we offer detailed comments on the specific references the reviewer mentioned.
>
> ---
> **Regarding short-term synaptic plasticity**, we agree this line of work is highly relevant. Fast Hebbian plasticity, where functional connectivity changes rapidly based on recent neural activity without permanent weight modification, is conceptually aligned with the working memory and ICL side of our framework. Attention mechanisms essentially implement a form of fast, input-dependent modulation closely related to Fast Weight Programmers (FWPs), with a separation between slow stable weights and fast input-dependent connections (we cite a relevant work in our paper line 243 right, on FWPs). That said, in current transformer architectures this modulation occurs only through attention while FFN modules remain effectively non-plastic, and short-term synaptic plasticity could be viewed as an inspiring design direction for richer working memory mechanisms going beyond standard attention. We will incorporate this line of work in the revised manuscript.
>
> **Regarding behavioral timescale plasticity**, we thank the reviewer for this pointer, this is an interesting active line of research that we were not fully aware of. At a high level, it touches on how weight updates (credit assignment) are governed, which connects to a broader and important direction of rethinking the learning algorithm itself for continual learning. An example of this direction in the CL literature is Continual Backprop, which modifies the update rule to tackle loss of plasticity. We view biologically inspired learning algorithms of this kind as a promising,  complementary direction and source of inspiration for works on core model consolidation.
>
> **Regarding low-rank RNNs**,  we believe this is particularly relevant to our discussion of consolidation and architectural/representational modularity (see Response to Reviewer *rg3q*). The finding that rank-based constraints enable generalization and task specialization within the network speaks directly to our argument on enforcing modularity and specialization within the core model and it is a promising direction for mitigating forgetting during consolidation. We will incorporate this as a concrete neuroscience-grounded motivation for the architectural ingredient of our consolidation discussion.

---

> > ### Author Rebuttal · Reviewer_FSdX · 2026-04-01
> >
> > As noted in my original review, this work is well within my own expertise, albeit my core field is theoretical neuroscience. The fact that I am quite excited about this work is testament that the position paper has made impact outside of its immediate field. I went through the rebuttal and remain comfortable recommending a strong accept.
> >
> > For authors, please also refer to my comment about "memory consolidation in the brain and hippocampal replay". Though this comment remains unanswered, it is not authors' fault. It seems I forgot to add it as a separate comment to address but mistakenly left it at the summary. I recommend authors take a look at related works and add a few sentences connecting their framework to these biological findings. Otherwise, all responses to my questions are satisfactory.

---

### Official Review · Reviewer_ap7R · 2026-03-13

**Significance:** 3
**Argument Clarity:** 4
**Rating:** 5
**Confidence:** 4

**Questions:**

1. What empirical result would most directly support the paper’s central thesis, and what kind of finding would count against it? At the moment the vision is compelling, but it would help to know what a decisive evaluation would actually look like.

2. Among the many design options presented, which ones do the authors see as the most important in practice? If someone were to build a first serious system based on this paper, what choices would matter most?

3. What form should the offline update process take? More specifically, how should information from longer-term storage influence the base model without causing the sort of instability the paper criticises in standard continual learning?

4. How often should this offline process run, and what should trigger it? This seems especially important if the framework is meant to support real-world agents rather than idealised ones with abundant compute.

**Alternative Views Section:**

Yes

**Compliance With Llm Reviewing Policy A Conservative:**

Affirmed.

**Discussion Potential:**

3

**Final Justification:**

After reading the paper, the other reviews, and the author's responses, I am maintaining my score of 5 (Accept).

My initial assessment highlighted this as a clearly written and well-organised synthesis that takes a timely and important position on combining in-weight and in-context learning under a modular memory framework, with a particularly strong treatment of alternative views. My main reservations concerned the underspecified consolidation story, the limited discussion of practical costs of maintaining multiple memory modules, and a missing connection to recent empirical work on episodic memory in LLMs. The rebuttal addressed these concerns substantively. In particular, I think the response to Reviewer rg3q added valuable specificity on consolidation through compressed memory representations, architectural modularity and parameter specialisation, RL-style training objectives that favour generalisation over memorisation, and biologically inspired triggers based on temporally structured replay. In addition, the authors also made a reasonable efficiency argument for combining ICL and IWL over the long term, and committed to incorporating the suggested references on human-inspired episodic memory.

I would still encourage the authors to discuss coordination overhead more directly and to strengthen the links to hippocampal replay, as Reviewer FSdX also noted. Overall, given that this is a position paper whose value lies in identifying an important problem, synthesising relevant literature, and pointing the community towards a promising direction, the discussion among reviewers reinforced my positive view, and I remain confident in recommending acceptance.

**Paper Summary:**

This paper argues that continual learning in AI agents requires the separation of memory into distinct components, rather than dependence on a single mechanism. The central argument is that fast adaptation during interaction and slower, more stable learning processes should be managed differently. The authors present a system comprising three elements: (a) a base model that encodes durable, general competence within its parameters; (b) a short-term store for immediately relevant information; and (c) a longer-term memory for experiences, facts, and accumulated knowledge. In this system, context-based adaptation addresses fast timescales, while parameter updates are reserved for selective, offline periods.

The paper also maps out the design space in a useful way. It compares different kinds of memory representation, discusses retrieval and storage choices, and considers questions such as when information should be retained, removed, or absorbed into the model itself. The discussion is informed by ideas from continual learning, cognitive science, and computer systems, and the paper closes by considering several application areas as well as competing viewpoints.

**Position:**

Yes

**Position In Title:**

Yes

**Related Work:**

4

**Strengths And Weaknesses:**

The paper is clearly written and well-organised. The authors explain their main idea effectively, and the distinction they draw between active use and offline updates strengthens their argument. The topic is timely, as there is a strong need for methods that go beyond simply enlarging context windows or repeatedly fine-tuning the same model. The unification of these concepts is therefore highly valuable.

Another strength is how the paper organizes ideas for the field. Breaking down memory types and comparing their pros and cons will help other researchers, especially those thinking about real-world design choices. The connections to complementary learning systems and hardware-like memory structures are also helpful. They put the proposal in context but without making it seem too speculative or abstract.

I also thought the discussion of alternative positions was handled well. The paper does not ignore the obvious objections, such as the claim that sufficiently large context windows plus retrieval are enough, or the opposing claim that continual parameter updates alone will solve the problem. It takes these alternatives seriously and responds to them in a technically informed way.

My primary reservation is that the paper is more convincing as a synthesis and agenda-setting piece than as a sharply testable position. The broad claim that a memory system with separate components is needed will probably not surprise many people working in this area. The more difficult and interesting question is which specific design decisions matter most, and here the paper remains somewhat non-committal. It lays out many possibilities, but is less decisive about which ones are most promising or which should be prioritised first.

A related issue is that the consolidation story still feels underspecified. The paper argues, reasonably, that knowledge should move more gradually into the core model, but it does not say enough about how this can happen without reintroducing the same interference problems that continual learning has struggled with for years. Since this transition is such a central part of the framework, I would have liked a clearer sense of what concrete mechanisms the authors believe are viable.

I also think the paper should discuss practical costs more. Maintaining several memory systems, deciding what moves between them, and running regular consolidation steps might be useful, but could also be a lot of work to build and maintain. This is especially important for agents with limited computing power or energy. The paper mentions complexity in general, but I would have liked a more direct look at when the extra systems are really worth it.

Finally, on a minor related work note: while the paper effectively draws on human episodic memory to motivate its framework, the argument would be more complete if it briefly acknowledged recent empirical work in this space. For instance, mentioning existing working implementations of human-inspired episodic memory in LLMs (e.g., Fountas et al., 2025), alongside discussions of the limitations and current trajectories of such approaches (e.g., Dong et al., 2025; Huet et al., 2025), would help bridge the gap between the paper's cognitive theories and current machine learning practice.

Overall, I am positive about this paper. It is thoughtful, well organised, and likely to be useful to the community, especially because it brings together lines of work that are too often discussed separately. I do not think it fully resolves the hardest part of its proposal, i.e. how consolidation into model weights should be carried out safely and efficiently, and I would have welcomed a stronger set of concrete commitments. Even so, as a position paper it succeeds in defining an important research direction and should prompt worthwhile discussion.

#### References:
* Dong et al. (2025). "Towards large language models with human-like episodic memory." Trends in Cognitive Sciences.
* Fountas et al. (2025). "Human-inspired Episodic Memory for Infinite Context LLMs." ICLR.
* Huet et al. (2025). "Episodic Memories Generation and Evaluation Benchmark for Large Language Models."

**Support:**

3

---

> ### Author Rebuttal · Authors · 2026-03-31
>
> We thank the reviewer for their positive judgement and constructive feedback. We would like to elaborate on the commitment and specificity of our framework.
>
> ---
> **Answering Weaknesses**
> ---
> **The point we committed to most strongly is that CL systems must embrace both ICL and IWL under a modular rather than monolithic architecture**, and we are explicit throughout the paper, including in the Alternative Views section, about why relying solely on either is insufficient.
>
> **On mechanism specificity**: We refer the reviewer to our response to W1 of Reviewer *H5bG* for our rationale behind the framework's level of generality.
>
> That said, **we believe that our discussions do provide insights on promising directions.** On memory representations, we analyze current memory representations against a set of desiderata in Table 1, namely per item storage, capacity, modality, update speed, interference risk, selective forgetting, retrieval, and generalization, which stand as important axes for future instantiations and research works building on our position. During the discussion there, we argue that raw data carries fundamental limitations, particularly in multimodal and embodied settings (explicit reference in Section 6), pointing toward learned or latent memory representations as a more promising direction, achieving the best trade offs. On memory functions, we implicitly point toward learned policies as the strongest long term direction, a point made explicit in the “Controlling the Flow” paragraph of Section 3, where we argue that intrinsically learned metacognitive control is ultimately essential for autonomous CL agents.
>
> **On  Consolidation**: See response to reviewer *rg3q*.
>
> **On practical costs**: We agree that practical efficiency is crucial and emphasize that it is a central focus of our framework. In Section 4, when discussing memory representations, we specifically address their efficiency desiderata (storage cost, retrieval cost, and update speed), and the same applies to memory structure, which directly affects retrieval efficiency as memory grows over time. Moreover, as pointed out throughout the manuscript, we believe that, over long-term operation, combining ICL and IWL is more efficient than either alone. Long-context processing is costly in memory and compute, while continual full-model updates are expensive and unstable. Together, they strike a balance: ICL enables rapid adaptation, reducing frequent updates, and IWL periodically consolidates knowledge, limiting context growth and retrieval overhead.
>
> **On related work**: We thank the reviewer for these pointers. We agree that acknowledging recent empirical work on human-inspired episodic memory in LLMs would strengthen the connection between cognitive motivation and current machine learning practice, and also provide the reader with additional concrete implementations of specific memory components (i.e. episodic memory). We will incorporate the mentioned works in the revised manuscript.
>
> ---
> **Answering Questions**
> ---
> **Q1**: As we discuss in section 6, a popular direction against our thesis is that long-context + RAG with a frozen core model is sufficient for continuous operation at scale, i.e. that infinitely scaling context windows and engineering retrieval pipelines along with model scale can fully substitute parametric adaptation. In contrast, the most direct empirical support would be demonstrating that smaller models equipped with principled modular memory and consolidation can match or outperform much larger models that rely solely on massive context and engineered RAG, especially in the streaming and multimodal settings of Section 5, where the limits of ICL-only approaches are most evident. In long horizon tasks, we expect such smaller models to gain capability over time by transferring information from working memory to long term memory and eventually into the core model, while maintaining a bounded memory footprint through forgetting policies.
>
> **Q2**: As noted earlier, we believe our discussion already points to promising directions. **A first instantiation of our framework could be**: (1) using raw data for working memory, given its transient nature, and learned internal embeddings for long term memory, since they capture multimodal information, model relations between experiences, and support efficient retrieval without handcrafted structures (as argued in Table 1); (2) adopting learned policies for memory functions (what to store, when to retrieve, when to forget) as the strongest long term direction, with intrinsically learned metacognitive control as the ultimate goal (Section 3, “Controlling the Flow”); and (3) using self study style objectives and on policy training for consolidation, as these promote generalization over memorization and reduce forgetting compared to supervised fine tuning (Section 4; see also our response to Reviewer *rg3q*).
>
> **Q3 & Q4**: We refer the reviewer to our response to Reviewer *rg3q*.

---

> > ### Author Rebuttal · Reviewer_ap7R · 2026-04-02
> >
> > I thank the authors for their rebuttal. After reading both this and their responses to the other reviewers, I'm confident that my main concerns have been addressed satisfactorily.
> >
> > Regarding consolidation, the added detail in the response to Reviewer rg3q, including compressed memory representations, architectural modularity and parameter specialisation, and RL-style training objectives that prioritise generalisation over memorisation, brings in an important level of specificity that was missing from the original manuscript. The discussion of consolidation triggers, inspired by temporally structured replay during sleep, is also a valuable addition. I would encourage the authors to include these points clearly in the revised paper, since they substantially strengthen what was previously the least specified part of the framework.
> >
> > Regarding practical costs, I accept the authors' argument that combining ICL and IWL may be more efficient over long-term operation than relying on either mechanism alone. Even so, I would still appreciate a short discussion of the overhead involved in maintaining and coordinating multiple memory modules, especially for agents with limited resources.
> >
> > I also appreciate that the authors plan to include suggested references on empirical work on episodic memory in LLMs. I would also second Reviewer FSdX's suggestion to make the links to memory consolidation in the brain and hippocampal replay more explicit.
> >
> > More generally, after reading the full discussion, I feel it is important to point out that this submission should be judged as a position paper. Position papers are not meant to prescribe concrete mechanisms or provide empirical benchmarks. Their value lies in clearly identifying an important problem, bringing together the relevant literature, and guiding the community toward a promising research direction. In my opinion, this paper does all three well, and the rebuttals have only strengthened that view.
> >
> > I am keeping my score at 5 (Accept).

---

### Official Review · Reviewer_H5bG · 2026-03-13

**Significance:** 3
**Argument Clarity:** 2
**Rating:** 4
**Confidence:** 3

**Questions:**

1. The proposed modular memory framework is defined so broadly that its boundaries become very loose. In its current form, almost any system that combines a core model with some form of replay buffer, memory bank, or hierarchical RAG plus occasional parameter updates can be subsumed under this umbrella. Can you precisely articulate what is fundamentally new in their framework compared to these existing modular or hierarchical memory systems?
2. How is long-term memory represented, and what are the precise rules for writing and retrieving?
3. On which benchmark regimes do you expect modular memory architectures to strictly outperform strong baselines like long-context+RAG?

**Alternative Views Section:**

Yes

**Compliance With Llm Reviewing Policy A Conservative:**

Affirmed.

**Discussion Potential:**

3

**Final Justification:**

The author addressed my concerns in rebuttal. Therefore, I raise my score to the positive side.

**Paper Summary:**

This paper argues that current continual learning efforts are hamstrung by an over-reliance on in-weight learning (IWL) and, more recently, a naive dependence on in-context learning (ICL) via ever-growing memories and frozen models.

**Position:**

Yes

**Position In Title:**

Yes

**Related Work:**

2

**Strengths And Weaknesses:**

Strengths:
1. The paper provides a conceptually clean framework, decomposing into core model, working memory, and long-term memory, together with two operational regimes.
2. The authors make efforts to connect ideas from classic CL, modern LLM memory/RAG systems, human memory, and computer architectures, providing a valuable synthesis to highlight that CL is a systems-level memory problem.

Weaknesses:
1. The paper mainly unifies different individual elements into a modular memory pipeline. However, the concrete mechanisms are not specified, making it hard to distinguish the truly new proposals/designs.
2. The problem is not well formulated. Lack of defining what tasks or evaluation protocols. The community would find it difficult to operationalize the position without at least one rough evaluation scenario.
3. The authors seek to address the concept of modular memory as the missing piece for continual learning. Yet recent systems already instantiate something similar, like the hierarchical memory banks.

**Support:**

2

---

> ### Author Rebuttal · Authors · 2026-03-31
>
> We thank the reviewer for taking the time to read our manuscript and for their comments.
>
> ---
> **Answering Weaknesses**
> ---
> **W1**:  Section 3 discusses the full system architecture and operations, outlining how ICL and IWL can co-exist for fast adaptation and long term capability enhancement. Section 4 delves into memory representations and their trade-offs in Table 1, along with the key memory functions (update, retrieval, consolidation). **Our goal in this position paper is to provide a long-term framework for tackling the core challenges of continual learning, rather than committing to a specific mechanism that may quickly become obsolete. We guide the reader through the design space and identify fundamental research questions while remaining implementation-agnostic.**
>
> **W2:** **In section 5, we provide several tasks and evaluation scenarios**, as Reviewer *FSdX* kindly acknowledges. Specifically, we identify concrete domains such as personalization, streaming video and embodied agents. For each, we discuss, within the limits of paper length, (a) which challenges of that domain are addressed by modular memory and (b) which gaps in current approaches our framework targets, citing papers from which benchmarks can be derived; we will make these more explicit in the revision. Given the well‑defined problem of continual learning (CL) and its challenges, we focus on the proposed framework as a bridge between current directions, while referencing works that define CL.
>
> **W3:** Our argument is not that memory components are absent from the literature, but rather that a principled, holistic framework unifying both ICL and IWL under explicit memory modules **with well-defined operational regimes and distinct learning timescales is still missing**, and crucially needed as a reference point for tackling Continual Learning at scale.
>
> Hierarchical memory systems such as hierarchical RAG rely exclusively on ICL for adaptation and thus suffer from the limitations discussed in our paper (Section 6 as acknowledged by reviewers *rg3q* and *ap7R*). Moreover, Section 2 provides a literature review including recent works on LLM memory agents, and we also cite a recent survey on memory for LLM agents (line 247). Our central claim is that systems with a frozen core model (i.e., ICL alone) are insufficient for Continual Learning at scale. We kindly ask the reviewer to share specific references if there are particular works they believe we overlooked.
>
> ---
> **Answering Questions**
> ---
> **Q1**:  We refer to our responses to W1 and W3. The framework the reviewer describes here is exactly what we argue is insufficient:
>
> - **On replay buffers (traditional CL)**: A replay buffer corresponds to pure IWL: a static data store used only to retrain the model. In contrast, our long-term memory is a dynamic module that stores and consolidates experiences to rapidly condition the core model on past experiences via ICL, while also being consolidated into the core model infrequently to drive higher-level generalization and reduce reliance on retrieval.
>
> - **On memory banks and hierarchical RAG with occasional parameter updates**: We refer to our response to Reviewer rg3q which discusses a related work on this and showcases that such a simplified combination remains insufficient since it overlooks the crucial operational details: what to store & forget, how to retrieve, how to consolidate, how to control the flow. These are precisely the questions our framework puts at the center, and which existing systems often ignore.
>
> **Q2**: Section 3 describes what long-term memory does at a functional level: it stores facts, events, and personalized experiences that persist beyond the current context, supports retrieval into working memory for rapid ICL-based adaptation, and is periodically consolidated into the core model via IWL to **drive higher-level generalization**.
>
> **Designing such a module involves** three levels of choices, all discussed in Section 4: (1) representation format (raw data, embeddings, neural memories); (2) memory structure (lists, trees, graphs) and (3) writing, forgetting, and retrieval policies. Again, consistent with the position paper scope, we outline the design space rather than prescribing a single instantiation.
>
> **Q3**: We refer to our previous response to Weakness 2. There is strong evidence from the literature of the weaknesses of long-context + RAG in several related settings, for example:
> - Computational cost and performance degradation with growing context (Section 6);
> - ICL for reasoning-heavy tasks [1,2];
> - Multimodal ICL [3] & long video understanding [4].
>
> These are precisely the challenges where we expect our framework to tackle.
>
> ---
> [1] https://arxiv.org/abs/2411.07279
>
> [2] https://arxiv.org/abs/2602.02366
>
> [3] https://arxiv.org/abs/2510.18117
>
> [4] https://huggingface.co/blog/timescope-video-lmm-benchmark

---

> > ### Author Rebuttal · Reviewer_H5bG · 2026-04-03
> >
> > Thanks for the rebuttal. My concerns are addressed. I will raise my score.

---

### Decision · Program_Chairs · 2026-04-30

**Decision:**

Accept (spotlight)

**Comment:**

The review process provided critical insights into the system-level challenges of the proposal:
•	Conceptual vs. Concrete: Reviewers initially noted that the framework was defined "broadly" and lacked precise mechanisms for memory representation and retrieval rules. The authors clarified in the rebuttal that their contribution is a holistic, principled reference point for unifying ICL and IWL, rather than a single specific algorithm.
•	Biological Alignment: Reviewers pointed out that the argument would be more complete by explicitly acknowledging the Complementary Learning Systems (CLS) theory in neuroscience (Hippocampus/Neocortex interaction). The authors embraced this, noting that humans remain the strongest example of continuous adaptive intelligence through such modularity.
•	Evaluation Scenarios: A key insight from the discussion was the need for new benchmarking. The authors proposed that "smaller models with modular memory matching the performance of massive RAG-based models" would be the ultimate empirical proof of their position
•	Technological Feasibility: Discussion emerged regarding "Memory as a First-Class Citizen," comparing AI memory management to persistent memory in computer architecture.

In summary this paper serves as a timely and necessary intervention in the Continual Learning debate, effectively bridging the gap between traditional parametric methods and modern LLM capabilities, and serves as a focal point for discussion on the architectural future of adaptive AI agents.